# Role of Repeat PET/CT Imaging in Head and Neck Cancer Following Initial Incomplete PET/CT Response to Chemoradiation

**DOI:** 10.3390/cancers13061461

**Published:** 2021-03-23

**Authors:** Austin J. Iovoli, Mark K. Farrugia, Sung Jun Ma, Jon M. Chan, Michael R. Markiewicz, Ryan McSpadden, Kimberly E. Wooten, Vishal Gupta, Moni A. Kuriakose, Wesley L. Hicks, Anurag K. Singh

**Affiliations:** 1Department of Radiation Medicine, Roswell Park Comprehensive Cancer Center, Elm and Carlton Streets, Buffalo, NY 14263, USA; Austin.Iovoli@RoswellPark.org (A.J.I.); Mark.Farrugia@roswellpark.org (M.K.F.); SungJun.Ma@RoswellPark.org (S.J.M.); 2Department of Head and Neck Surgery, Roswell Park Comprehensive Cancer Center, Elm and Carlton Streets, Buffalo, NY 14263, USA; Jon.Chan@RoswellPark.org (J.M.C.); Michael.Markiewicz@RoswellPark.org (M.R.M.); Ryan.McSpadden@RoswellPark.org (R.M.); Kimberly.Wooten@RoswellPark.org (K.E.W.); Vishal.Gupta@RoswellPark.org (V.G.); Moni.Kuriakose@RoswellPark.org (M.A.K.); Wesley.Hicks@RoswellPark.org (W.L.H.J.)

**Keywords:** cancer of head and neck, PET/CT, imaging, surveillance, chemoradiation

## Abstract

**Simple Summary:**

Following completion of chemotherapy and radiation for the treatment of head and neck cancer, a PET/CT scan is typically obtained 3 months later to assess how well the patient responded to treatment. The results of this PET/CT are often difficult to interpret because radiation can cause inflammation around the area being treated that can take months to resolve. We looked at 57 patients who had a repeat PET/CT scan performed after initial post-treatment imaging was unclear to examine whether this was helpful in determining whether these patients require further testing. Among this group, 48% of patients converted to having a complete response to treatment and none went on to develop treatment failure. Based on our findings, repeat PET/CT imaging can provide valuable information for head and neck cancer patients that can reduce the incidence of unnecessary biopsies and surgeries.

**Abstract:**

Despite waiting 13 weeks to perform a PET/CT scan after completion of chemoradiation for head and neck squamous cell carcinoma (HNSCC), equivocal findings are often found that make assessing treatment response difficult. This retrospective study examines the utility of a repeat PET/CT scan in HNSCC patients following an incomplete response on initial post-treatment imaging. For this cohort of 350 patients, initial PET/CT was performed 13 weeks after completion of treatment. For select patients with an incomplete response, repeat PET/CT was performed a median of 91 days later. Primary endpoints were conversion rate to complete response (CR) and the predictive values of repeat PET/CT imaging. Of 179 patients who did not have an initial complete response, 57 (32%) received a repeat PET/CT scan. Among these patients, 26 of 57 (48%) had a CR on repeat PET/CT. In patients with CR conversion, there were no cases of disease relapse. The sensitivity, specificity, PPV, and NPV for the repeat PET/CT for locoregional disease were 100%, 59%, 42%, and 100%. Repeat PET/CT in HNSCC patients with an incomplete post-treatment scan can be valuable in obtaining diagnostic clarity. This can reduce the incidence of unnecessary biopsies and neck dissections.

## 1. Introduction

Routine surveillance is practiced in head and neck squamous cell carcinoma (HNSCC) patients with the objective of identifying early recurrent or persistent disease. After completion of treatment, fluorodeoxyglucose positron emission tomography with diagnostic computed tomography (PET/CT) scan 3–6 months later is often performed to assess disease response due to its high sensitivity and negative predictive value [1,2,3,4,5,6,7,8]. The timing of PET/CT imaging at least 12 weeks after treatment completion is thought to limit false-positive findings by allowing resolution of post-surgical and radiation-induced changes in normal tissue that may interfere with detection of persistence against the need to identify treatment failure early in order to increase the chance of cure with salvage therapy [9]. These post-treatment PET/CT findings have diagnostic importance and provide clinically useful prognostic information to help guide further management [10,11]. Initial 12 week post-treatment PET/CT surveillance has been shown to not only offer comparable survival compared to patients undergoing planned neck dissection but also be more cost-effective [12,13]. Current National Comprehensive Cancer Network guidelines recommend assessment of extent of disease after completion of treatment with PET/CT at a minimum of 12 weeks as a preferred approach while the United Kingdom National Multidisciplinary Guidelines recommend post-therapeutic imaging with PET/CT only to be performed when recurrence is suspected [14,15].

Several studies of serial post-treatment PET/CT imaging found limited benefit from routine use of subsequent PET/CT surveillance, particularly when initial imaging was negative [16,17]. In light of this, other investigators have examined the use of a repeat post-treatment PET/CT in select patients who had an incomplete response on initial imaging and found that this can provide diagnostic clarity [18,19]. These prior studies focused predominantly on human papilloma virus-associated (HPV+) oropharyngeal carcinoma and the utility of this approach in a heterogenous HNSCC population that includes HPV- patients remains unknown. This retrospective study was performed using a large single-institutional HNSCC database to further examine the utility of performing a repeat PET/CT scan in patients with equivocal post-treatment imaging, with the goal of using repeat imaging to spare patients unnecessary surgical intervention.

## 2. Materials and Methods

Our Institutional Review Board approved this retrospective study of non-metastatic HNSCC patients diagnosed and treated with definitive radiation (RT) or chemoradiation (CRT) between 2007 and 2017 (EDR-103707). 

### 2.1. Eligibility

To be included in this study, the primary tumor had to be: (1) an invasive squamous cell carcinoma limited to the head and neck, (2) treated with definitive RT or CRT, and (3) successfully completed treatment. Patients were excluded if they had a previous history of non-cutaneous malignancy or if response to treatment was unable to be evaluated. Our complete patient selection criteria are shown in Figure 1. Overall, 831 primary HNSCC patients were diagnosed or treated between 2007 and 2017. Of these, 350 (42%) patients met the above selection criteria and had complete follow up data. 

### 2.2. Data Collection, Treatment, and Follow Up 

Demographic and clinical characteristics of study subjects, including age, gender, social habits, comorbidities (respiratory, cardiovascular, immune, renal, endocrine), and previous cancer history, were collected by a detailed medical chart review. Clinical information such as stage, grade, HPV status, anatomical subsite, treatment modality, cancer recurrence, survival status, survival duration, and cause of death were also collected. During the early years of the cohort, HPV testing was not routine. It was conducted on clinical request by in situ hybridization (HPV 16/18 biotinylated DNA probe Y1412; Dako, Carpinteria, CA, USA). In the later years of this study, HPV testing was routine and omitted only if there was insufficient tumor sample due to diagnosis by fine needle aspiration. P16 staining as a surrogate marker for HPV positivity has been routinely done for the several years. The cisplatin-based chemotherapy (weekly or every three weeks) and intensity modulated radiation therapy regimens (70 Gy to the primary tumor and 56 Gy to the elective lymph nodes in 35 fractions) applied in this cohort have been previously described in detail [20,21].

Following completion of chemoradiation, patients were seen in follow up for clinical exam at 4 weeks and 3 months. Further follow up was performed every 3 months in the first year, every 4 months for the next 2 years, and every 6 months for the next 2 years. Follow up visits alternated between surgical oncology and radiation oncology to reduce number of patient appointments.

### 2.3. Treatment Response Assessment with PET/CT Imaging

FDG PET/CT was performed on all patients at 13 weeks following completion of chemoradiation to assess response to treatment. PET/CT imaging was evaluated by radiologists working at a dedicated cancer center and board certified in nuclear medicine. The radiologist report of each PET/CT scan was qualitatively assessed by a single investigator for radiographic treatment response as complete, equivocal, persistent, or metastatic. Findings were considered a complete response if there was no residual FDG avidity above background or diffuse uptake without a corresponding structural abnormality that appears suspicious. In cases where FDG avidity was of greater intensity than nearby normal tissue activity but below background liver activity, findings were considered equivocal. Equivocal responses were further subclassified as equivocal due to suspicious locoregional and/or distant findings. When FDG avidity was focal, greater intensity than background liver uptake, and corresponded to a structural abnormality it was considered persistent if located in the head and neck region or metastatic if located at a site of distant spread. Patients who remained clinically equivocal after multidisciplinary review of post-treatment imaging and clinical exam findings were ordered a repeat PET/CT, typically 1–4 months following the patient’s initial post-treatment imaging. Repeat PET/CT imaging was assessed in the same way as initial post-treatment imaging.

### 2.4. Statistical Analyses

The first primary endpoint was conversion rate of initial incomplete locoregional response to complete response (CR) on repeat PET/CT. The second primary endpoint was evaluating the predictive values of initial and repeat post-treatment PET/CT imaging. For the purpose of analysis, persistent and equivocal locoregional responses were combined as positive imaging findings. Clinical disease status was determined by review of all subsequent histopathological and radiographic reports following completion of treatment. 2 × 2 tables were assembled with patient clinical outcomes and used to calculate sensitivity, specificity, positive predictive value (PPV), and negative predictive value (NPV). Survival trends for overall survival (OS), disease-specific survival (DSS), and freedom from progression (FFP) were estimated using Kaplan–Meier survival curves. All tests are two sided and performed at a nominal significance level of 0.05. SAS version 9.4 (SAS Institute, Cary, NC, USA) and R version 4.0.2 (R Project for Statistical Computing, Vienna, Austria) were used for statistical analyses.

## 3. Results

A total of 350 patients were included in the analysis. Baseline patient, tumor, and treatment characteristics for the total cohort and those receiving a repeat PET/CT scan are described in Table 1. Median age for the cohort was 58 years (range 18–90) and 84% of patients were male. Chemotherapy was a component of treatment in 96% of patients. Median follow up was 36 months (interquartile range (IQR) 21–67 months). For the entire cohort, HPV evaluation was positive in 145 (41%), negative in 73 (21%) and unknown in 132 (37%).

Initial post-treatment PET/CT was performed a median of 91 days (IQR 90–96) after completion of treatment. Assessment of initial PET/CT is described in Figure 2 and included 171 complete (49%), 132 equivocal (38%), 33 persistent (9%), and 14 metastatic responses (4%). Among complete responders, 6/171 (4%) developed local failure, 6/171 (4%) developed regional failure, 2/171 (1%) developed both local and regional failure, 15/171 (9%) developed distant failure, and 2/171 (1%) developed both locoregional and distant failure. Of the 14 patients with radiographic evidence of metastatic disease, 9 (64%) pursued palliative care, 2 (14%) went on systemic therapy, and 1 (7%) was successfully salvaged for a solitary lung metastasis. Between those who achieved a complete vs. equivocal response on initial PET/CT imaging, 3 year FFP (82.8% vs. 68.6%, *p* = 0.0018), 3 year DSS (89.6% vs. 73.0%, *p* < 0.0001), and 3 year OS (88.6% vs. 71.0%, *p* < 0.0001) were all significantly improved in the complete responders (Figure 3). The sensitivity, specificity, PPV, and NPV for locoregional and distant disease on initial post-treatment PET/CT were 75%, 59%, 43%, and 85% (Table 2). When further stratified in those with known HPV status, the sensitivity, specificity, PPV, and NPV for locoregional and distant disease on initial post-treatment PET/CT were 66%, 65%, 37%, and 86% for HPV-positive cancers (*n* = 147) and 68%, 54%, 50%, and 71% for HPV-negative cancers (*n* = 84).

Of 179 patients who did not have an initial complete response, 57 (32%) received a repeat PET/CT scan a mean of 90 days and median of 91 (IQR 70–98) days later. Among the 57 patients who received a second post-therapeutic staging PET/CT scan, 2 were performed under 40 days and 7 occurred under 60 days from initial imaging. Assessment of repeat PET/CT imaging and subsequent clinical outcomes are described in Figure 4. Among patients who received repeat imaging, 26 of 57 (48%) had a CR conversion. None of the patients who achieved CR conversion went on to have a locoregional or distant relapse. When stratified by those with known HPV status, 12 of 18 (67%) HPV-positive cancers and 4 of 12 (33%) HPV-negative cancers had a CR conversion. Shown in Figure 5, patients who had a complete response on repeat vs. initial PET/CT imaging had improved 3 year FFP (100% vs. 82.8%, *p* = 0.022) and similar 3 year DSS (100% vs. 89.6%, *p* = 0.091) and 3 year OS (95.8% vs. 88.6%, *p* = 0.50).

Patients with an incomplete response had equivocal (15 patients, 48%), persistent (13 patients, 42%), or metastatic (3 patients, 10%) findings on their repeat PET/CT. Among the 15 patients with equivocal findings, 9 patients went on to receive routine clinical follow up and 6 patients received further workup. Among the 13 patients with persistent findings, 3 patients went on to receive routine clinical follow up and 8 patients received further workup. For those receiving further workup, 9 of 14 (70%) patients were positive for recurrent or persistent disease. The sensitivity, specificity, PPV, and NPV for locoregional disease on the repeat PET/CT were 100%, 59%, 42%, and 100% (Table 3). When further stratified in those with known HPV status, there were no locoregional failures in patients with HPV-positive cancer (*n* = 18) and thus predictive values cannot be calculated. For those with HPV-negative cancer (*n* = 12), the sensitivity, specificity, PPV, and NPV for locoregional disease on repeat PET/CT were 100%, 57%, 63%, and 100%.

## 4. Discussion

This retrospective study demonstrated that a repeat PET/CT scan can provide diagnostic clarity in HNSCC patients with equivocal findings on initial post-treatment PET/CT imaging. In our cohort, 48% of patients who received repeat imaging had radiographic conversion to CR and an additional 19% of patients who did not achieve CR were felt after further clinical assessment to be free of disease without requiring additional workup. This was supported by a high NPV of 100% for repeat imaging, which helped stratify patients with the highest likelihood of having persistent disease to go on and receive further workup. Furthermore, while OS, DDS, and FFP were significantly increased in those achieving a complete response over those with an equivocal response on initial PET/CT imaging, patients with conversion to CR upon repeat imaging had similar survival to those initially achieving a complete response.

Our study is the first to examine repeat PET/CT imaging in a heterogenous cohort stratified by HPV status that includes nearly half non-oropharynx patients. In patients with known HPV status, we found that 67% of HPV-positive patients achieved a CR compared to 33% of HPV-negative patients. In a series of 562 HNSCC patients made up primarily of oropharyngeal carcinoma (85%), Prestwich et al. found that among 40 patients receiving a second-look PET/CT scan, the locoregional CR rate was 60% [19], which is comparable to our overall CR rate of 48%. They also similarly demonstrated a high NPV of 95% for the primary site and 100% for lymph nodes for repeat PET/CT imaging [19]. Based on these results, their current institutional practice is to recommend a neck dissection if equivocal or positive nodal findings fail to convert to CR on repeat PET/CT imaging. In another series limited to node positive HPV+ oropharyngeal HNSCC, Liu et al. examined the utility of a repeat PET/CT scan to help direct neck management in patients with an incomplete nodal response. They found that in 41 patients who underwent repeat PET/CT 4 weeks after initial post-treatment imaging, the NPV was high at 97% and 71% converted to nodal CR [18]. This is consistent with the CR rate found in our study for HPV+ patients. Vainshtein et al. also found the use of additional PET/CT surveillance in HPV+ oropharyngeal HNSCC resulted in substantially fewer unnecessary neck dissections than would have been indicated by CT imaging alone [22]. A separate study examining HPV+ oropharyngeal HNSCC found that 79% of equivocal responders on initial 12 week PET/CT imaging had no active disease in the neck at 6 months, suggesting a period of longer surveillance should be considered before making decisions regarding a neck dissection [23]. These studies support our findings that repeat PET/CT imaging, particularly in HPV+ patients, can guide patient selection for further workup to help avoid surgical intervention in those who do not require it.

There were no other studies that looked at repeat PET/CT imaging specifically in an HPV-negative patient population. While our study was limited by a smaller number of patients with known HPV-negative status, the CR rate we found was half that of HPV-positive patients. For both initial and repeat PET/CT imaging the PPV was greater for HPV-negative patients relative to HPV-positive patients as well. This difference could be partially attributed to the lower pre-treatment probability that HPV-positive disease will have residual disease [18]. Given these findings clinicians should consider a lower threshold for working up HPV-negative patients for residual disease when equivocal findings are found on post-treatment imaging.

While repeat PET/CT imaging in selected patients with equivocal findings may provide benefit, prior studies have demonstrated that the routine use of PET/CT scans outside the 3–6 month post-treatment window does not provide as much utility [16,17]. Perie et al. looked at systematically performing a PET/CT scan 1 year after treatment completion for patients with HNSCC and found that this was scarcely useful in patients without clinical suspicion of disease [17]. Similarly Ho et al. found limited benefit to serial PET/CT imaging at 12 and 24 months after completion of treatment for HNSCC patients with negative 3 month PET/CT imaging [16]. These results indicate that careful selection of patients for repeat PET/CT imaging should be employed to maximize benefit. While the Lui et al. study used a systematic approach to patient selection, our policy and that used by Prestwich et al. of limiting repeat imaging to those felt to benefit upon multi-disciplinary discussion yielded similarly positive results [18,19]. 

Interestingly, those who had a CR conversion on repeat imaging had improved FFP when compared to patients with CR on initial PET/CT, with trends observed towards better DSS as well. Given the large difference in sample number and the fact those with a CR conversion by definition avoided metastatic failure at approximately 6 months versus 3 months, it may be improper to conclude that a CR conversion is more favorable. Rather, these findings support the notion that a delayed CR is likely no different prognostically then an initial CR on PET/CT imaging. 

Based on our current results, we have changed our institutional policy to obtain an initial PET/CT scan 17 weeks after completion of treatment for HNSCC to allow more time for resolution of radiation-induced changes on normal tissue. The need for repeat PET/CT imaging in patients who continue to have a clinically and radiographically equivocal treatment response will be monitored and hopefully will decrease.

There are several limitations to this manuscript. Due to the nature of a retrospective study, the results are prone to information bias from miscoding of patient data and medication entry errors. In addition, there was not a standard protocol for selection of patients for repeat PET/CT imaging, rather this was determined by multidisciplinary discussion based on radiographic and clinical factors pertaining to each patient. This created variation in the timeframe patients received a repeat PET/CT as this was at the treating physicians’ discretion. Another limitation is the lack of a standardized reporting criteria used to interpret post-treatment PET/CT imaging for our cohort. The American College of Radiology has introduced a Neck Imaging Reporting and Data Systems (NI-RADS) protocol to provide radiologists a standard template to report surveillance imaging findings in HNSCC, with the management recommendation for NI-RADS 2 findings being direct inspection for mucosal abnormalities or short-interval follow up with CT or an additional PET for deep abnormalities [24]. A comparison between different standardized interpretive criteria for HNSCC PET/CT imaging, including NI-RADS, found similar diagnostic performance characteristics between each protocol [25]. Further investigation to establish a consensus recommendation for a validated standardized interpretive criteria in this population would be a key resource to help clinicians guide management in equivocal cases. 

## 5. Conclusions

Repeat PET/CT in HNSCC patients with an incomplete post-treatment scan can be valuable in obtaining diagnostic clarity. This can reduce the incidence of unnecessary biopsies and neck dissections.

## Figures and Tables

**Figure 1 cancers-13-01461-f001:**
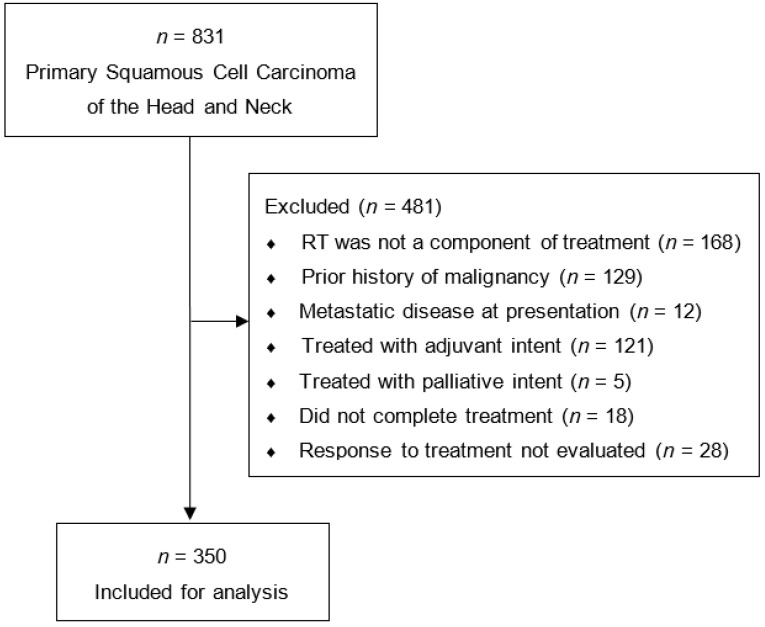
Patient selection criteria.

**Figure 2 cancers-13-01461-f002:**
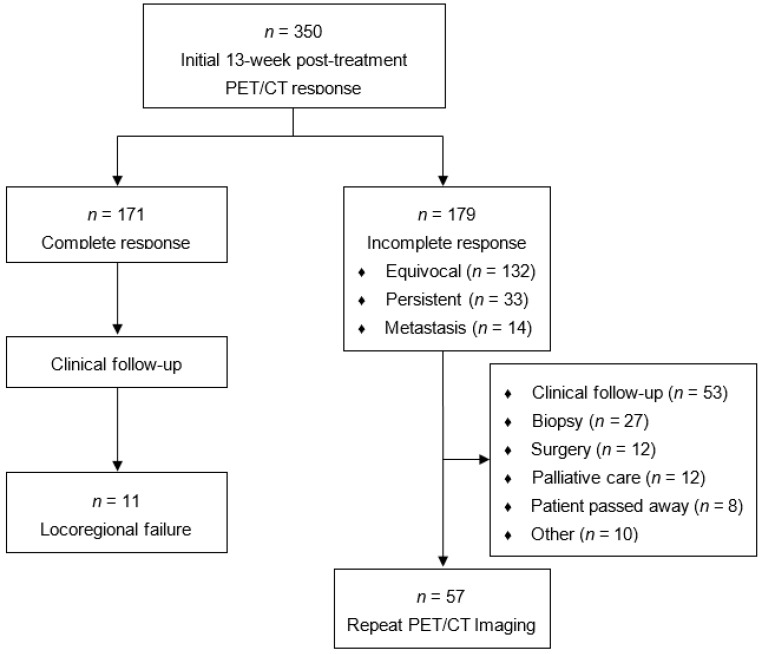
Flowchart of patient responses and clinical outcomes following initial 13 week post-treatment PET/CT scan.

**Figure 3 cancers-13-01461-f003:**
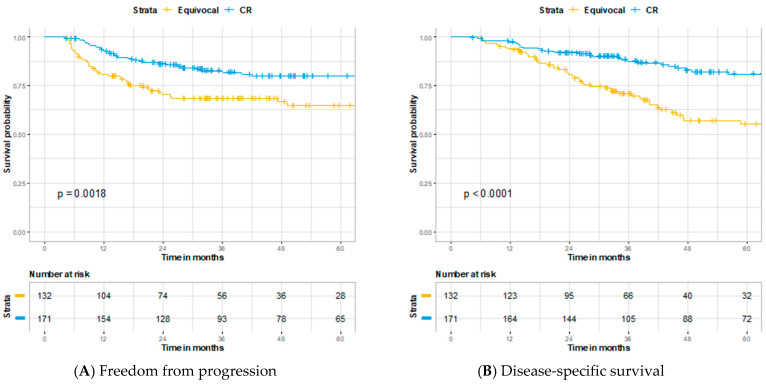
Kaplan–Meier curves comparing patients following initial PET/CT imaging with a complete response versus an equivocal response: (**A**) Freedom from progression, (**B**) disease-specific survival, and (**C**) overall survival

**Figure 4 cancers-13-01461-f004:**
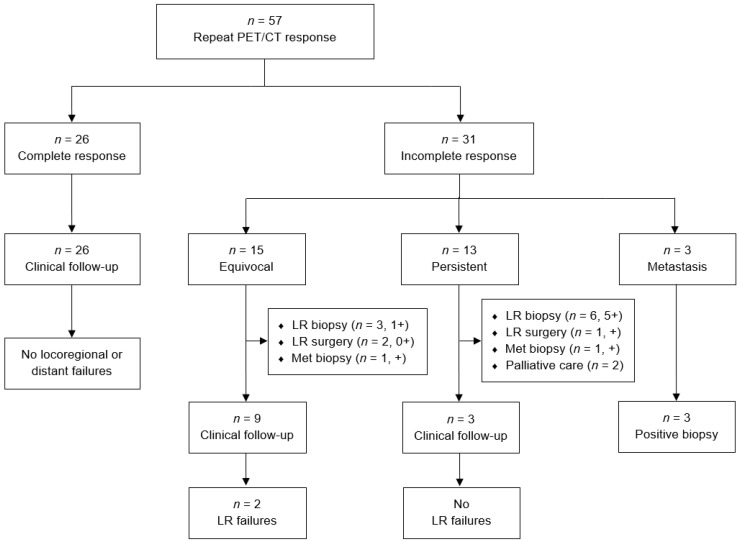
Flowchart of patient responses and clinical outcomes following repeat PET/CT scan.

**Figure 5 cancers-13-01461-f005:**
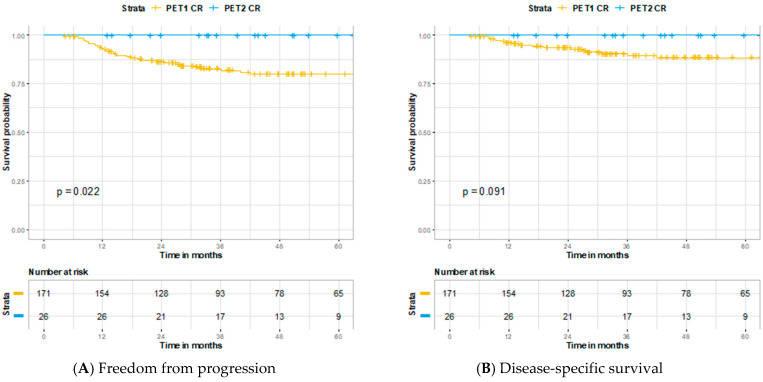
Kaplan–Meier curves comparing patients who achieved a complete response on initial PET/CT imaging versus those who achieved complete response on repeat PET/CT imaging: (**A**) Freedom from progression, (**B**) disease-specific survival, and (**C**) overall survival

**Table 1 cancers-13-01461-t001:** Baseline patient, tumor, and radiation treatment characteristics for the total cohort and patients who underwent repeat PET/CT.

	Total Cohort (*n* = 350)	Repeat PET/CT (*n* = 57)
Age (y)		
Median +/−SD	58 ± 9.6	56 ± 10.4
Range	18–90	18–88
Gender, *n* (%)		
Male	295 (84.3)	48 (84.2)
Female	55 (15.7)	9 (15.8)
Smoking, *n* (%)		
Never	84 (24.0)	10 (17.5)
Former	174 (49.7)	30 (52.6)
Current	92 (26.3)	17 (29.8)
T Stage *, *n* (%)		
T0	23 (6.6)	3 (5.3)
Tis	1 (0.3)	1 (1.8)
T1	46 (13.1)	6 (10.5)
T2	107 (30.6)	14 (24.6)
T3	123 (35.1)	22 (38.6)
T4	50 (14.3)	11 (19.3)
N Stage *, *n* (%)		
N0	77 (22.0)	17 (29.8)
N1	39 (11.1)	1 (1.8)
N2	199 (56.9)	31 (54.5)
N3	35 (10.0)	8 (14.0)
Overall Clinical Stage *, *n* (%)		
I	2 (0.6)	0 (0)
II	23 (6.6)	7 (12.3)
III	73 (20.9)	6 (10.5)
IVA	219 (62.6)	38 (66.7)
IVB	33 (9.4)	6 (15.8)
Primary Tumor Site, *n* (%)		
Nasopharynx	13 (3.7)	4 (7.0)
Oropharynx	187 (53.4)	27 (47.4)
Oral cavity	10 (2.9)	2 (3.5)
Larynx	85 (24.3)	16 (28.1)
Hypopharynx	32 (9.1)	5 (8.8)
Unknown primary	23 (6.6)	3 (5.3)
HPV Status, *n* (%)		
Negative	73 (20.9)	12 (21.1)
Positive	145 (41.4)	18 (31.6)
Unknown	132 (37.7)	27 (47.4)
Treatment, *n* (%)		
RT alone	13 (3.7)	2 (3.5)
ICT + CCRT	38 (10.9)	7 (12.3)
CCRT	299 (85.4)	48 (84.2)
Relapses, *n* (%)		
Local	41 (11.7)	11 (19.3)
Regional	26 (7.4)	2 (3.5)
Distant	62 (17.7)	7 (12.3)
Deaths, *n* (%)		
Total	128 (36.6)	23 (40.4)

* American Joint Committee on Cancer 7th edition.

**Table 2 cancers-13-01461-t002:** Predictive values of initial post-treatment PET/CT for persistent locoregional and distant disease.

Locoregional Status	Disease Recurrence	Disease Controlled	-
Initial PET CR	26	145	NPV: 85%
Initial PET IR	77	102	PPV: 43%
-	Sensitivity: 75%	Specificity: 59%	-

**Table 3 cancers-13-01461-t003:** Predictive values of repeat post-treatment PET/CT for persistent locoregional disease.

Locoregional Status	Disease Recurrence	Disease Controlled	-
Repeat PET CR	0	26	NPV: 100%
Repeat PET IR	13	18	PPV: 42%
-	Sensitivity: 100%	Specificity: 59%	-

## Data Availability

The data presented in this study are available on request from the corresponding author. The data are not publicly available due to privacy.

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
