# Peer review of "Role of Repeat PET/CT Imaging in Head and Neck Cancer Following Initial Incomplete PET/CT Response to Chemoradiation"

_cancers, 2021, doi:10.3390/cancers13061461_

Round 1

Reviewer 1 Report

Thank you for the opportunity to review this manuscript. 

This is a well-written manuscript on an important topic of assessing treatment response following definitive head and neck radiotherapy. My suggestions are:

  1. Figure 3 and 5. Please check if DSS and OS charts are reversed. DSS should have fewer events (deaths from cancer) compared to OS (deaths from any cause).
  2. Table 2. Please check the labels of Locoregional Status. Should they be "Initial" rather than "Repeat"?
  3.  The significance of HPV is brought up at several points of the manuscript and so the results on sensitivity, specificity, NPV, PPV should also include an analysis by HPV status.

Reviewer 2 Report

The study focuses on an interesting and important topic and underlines the value of PET/CT imaging in patients with head and neck cancer who underwent chemoradiation.

The following points should be fixed by the authors:

  • The authors use the term "review" several times. Since this is supposed to be an original article, this term should be avoided.
  • The authors don't provide any information about guidelines for imaging in head and neck cancer patients. Please add if PET/CT is recommended and if so, when/how often. References should be added.
  • In Figure 1 and 2, the text in the bottom line inside the box often is cut off. Please fix.
  • Authors only cite 18 references and several reviews. Key publications should be cited separately.
  • You mention that previous studies with larger patient cohorts showed similar results. Therefore, the diagnostic value of repeat PET/CT imaging after chemoradiation is not new. What is the novelty of your study? You should elaborate this in the Discussion.
  • You mention that the second posttherapeutic staging was conducted 1-4 months after chemoradiation was completed.  Were there many patients who had the second scan only one month after the first? What was the mean time between the 1st and the 2nd scan?

Round 2

Reviewer 2 Report

Thank you for your nice revisions. This is an interesting and well-written article and is now suitable for publication.